# Progression of Non-Significant Mitral and Tricuspid Regurgitation after Surgical Aortic Valve Replacement for Aortic Regurgitation

**DOI:** 10.3390/jcm12196280

**Published:** 2023-09-29

**Authors:** Shirit Kazum, Mordehay Vaturi, Idit Yedidya, Shmuel Schwartzenberg, Olga Morelli, Keren Skalsky, Hadas Ofek, Ram Sharony, Ran Kornowski, Yaron Shapira, Alon Shechter

**Affiliations:** 1Department of Cardiology, Rabin Medical Center, Petach Tikva 4941492, Israel; 2Faculty of Medicine, Tel Aviv University, Tel Aviv 6997801, Israel; 3Department of Thoracic Heart Surgery, Rabin Medical Center, Petach Tikva 4941492, Israel; 4Department of Cardiology, Smidt Heart Institute, Cedars-Sinai Medical Center, Los Angeles, CA 90048, USA

**Keywords:** mitral regurgitation, tricuspid regurgitation, surgical aortic valve replacement, aortic regurgitation, progression

## Abstract

Little is known about the natural history of non-significant mitral and tricuspid regurgitation (MR and TR) following surgical aortic valve replacement (SAVR) for aortic regurgitation (AR). We retrospectively analyzed 184 patients (median age 64 (IQR, 55–74) years, 76.6% males) who underwent SAVR for AR. Subjects with significant non-aortic valvulopathies, prior/concomitant valvular interventions, or congenital heart disease were excluded. The cohort was evaluated for MR/TR progression and, based on the latter’s occurrence, for echocardiographic and clinical indices of heart failure and mortality. By 5.8 (IQR, 2.8–11.0) years post-intervention, moderate or severe MR occurred in 20 (10.9%) patients, moderate or severe TR in 25 (13.5%), and either of the two in 36 (19.6%). Patients who developed moderate or severe MR/TR displayed greater biventricular disfunction and functional limitation and were less likely to be alive at 7.0 (IQR, 3.4–12.1) years compared to those who did not (47.2 vs. 79.7%, *p* < 0.001). The emergence of significant MR/TR was associated with preoperative atrial fibrillation/flutter, symptomatic heart failure, and above-mild MR/TR as well as concomitant composite graft use, but not with baseline echocardiographic measures of biventricular function and dimensions, aortic valve morphology, or procedural aspects. In conclusion, among patients undergoing SAVR for AR, significant MR/TR developed in one fifth by six years, correlated with more adverse course, and was anticipated by baseline clinical and echocardiographic variables.

## 1. Introduction

Aortic regurgitation (AR) is a condition in which the aortic valve (AV) fails to prevent systemic blood from back flowing into left ventricle (LV) during diastole. Constituting one of the most common valvular diseases in adults worldwide, AR usually manifests as a slowly progressive disease characterized by a gradual increase in LV volume load, a compensatory rise in chamber size and mass (i.e., eccentric remodeling and hypertrophy), and finally biventricular malfunction—ultimately translating to clinical heart failure (HF). At present, the definite treatment of unrepairable significant AR accompanied by signs or symptoms of cardiac dysfunction is surgical AV replacement (SAVR). While possible, addressing additional valvulopathies at the time of operation comes at the price of a lengthier procedure, potential complications, and higher costs, all of which could outweigh any theoretical benefit of a one-time multivalvular intervention. Yet, considering the worse prognosis and increased mortality associated with the co-presence of significant AR and mitral and/or tricuspid regurgitation (MR and/or TR) [1,2,3], current position papers and practice guidelines advocate the latter’s correction in parallel to SAVR [4,5,6]. Still, there is no consensus regarding the management of non-significant (i.e., less than moderate) MR and/or TR at the time of SAVR for AR, reflecting the paucity of data on the natural history of these valvular disorders in the context of AR, as most studies to date have focused on patients with either stenotic or mixed (rather than regurgitant) AV pathologies [7,8,9,10,11]. As a first step towards improving the decision-making process in this area of uncertainty, we examined the frequency of less than moderate MR or TR deterioration following SAVR for AR and further evaluated predictors for its occurrence, all using a large and contemporary database.

## 2. Materials and Methods

### 2.1. Study Population and Outcomes

Our study is based on the Rabin Medical Center registry of consecutive SAVR procedures performed for moderate-to-severe or severe AR on adult patients between 1 January 1996 and 31 December 2020. Included in the study were patients who exhibited less than moderate MR or TR at the baseline and for whom there was at least one retrievable transthoracic echocardiogram (TTE) prior to SAVR and two after it, one of them within the first six postprocedural months. We excluded patients with any of the following: 1. Greater than mild mitral or tricuspid stenosis; 2. Prior or concomitant non-aortic valvular interventions; 3. Concurrent LV assist device implantation; 4. Congenital heart disease; and 5. Acute intra- or postprocedural development of significant MR or TR due to a surgical complication.

The primary outcome was the incidence of MR or TR progression to moderate or severe on the last documented TTE. Based on the occurrence of this composite endpoint, the cohort was also retrospectively assessed for accompanying echocardiographic indices of ventricular and valvular function, New York Heart Association (NYHA) functional class at 1-year and at the last visit, and all-cause mortality along the entire follow-up period.

Conforming to the Declaration of Helsinki, the study was approved by Rabin’s Institutional Review Board (number 0603-23-RMC) which waived the need for informed consent.

### 2.2. Procedural Aspects

SAVR was undertaken following a dedicated heart team discussion that considered the best medical evidence at the time, practice guidelines [5,6], and patient preferences. Most procedures were performed via median sternotomy. Cardiopulmonary bypass was achieved by ascending aortic and double-stage venous cannulations, utilizing antegrade moderate hypothermic (28–30 degrees Celsius) cardioplegia. Actual valve replacement was performed according to standard pledged and interrupted-suture techniques. Transesophageal echocardiography and right heart catheterization were used for guidance, monitoring, and evaluation of the surgical result, as appropriate.

### 2.3. Echocardiographic Assessment

Echocardiograms at all stages were performed and interpreted by a team of experienced sonographers and level III-trained echocardiologists in accordance with accepted guidelines [12,13,14,15]. The echo machines used were Sonos-5500, Sonos-7500, IE-33, and EPIQ-7 (Philips, Andover, MA, USA) as well as Vivid-7 and Vivid-I (General Electric, Boston, MA, USA).

Regurgitation severity at all positions was determined in real-time by integration of qualitative (e.g., color Doppler-driven) and (semi)quantitative (e.g., spectral Doppler-derived) measures, whenever feasible, and graded as 0 (none-to-minimal), 1 (mild or mild-to-moderate), 2 (moderate), 3 (moderate-to-severe), or 4 (severe and greater). For the purpose of the study and in view of the guidelines, MR and/or TR of moderate, moderate-to-severe, and severe degrees were collectively referred to as “moderate or severe.” In cases of diagnostic ambiguity regarding AR extent, a multimodality approach was employed as deemed appropriate by the treating team, which utilized cardiac magnetic resonance and/or cardiac computed tomography for better volumetric assessment of regurgitant fraction and LV function and dimensions [16,17].

Global right ventricular (RV) function was assessed qualitatively and RV dilatation was defined as an end-diastolic RV diameter of 4.2 cm or greater by the apical 4-chamber view.

### 2.4. Data Collection

Echocardiographic parameters were retrieved from electronically stored reports, which were verified and amended as needed by a consensus of at least two echocardiologists taking part in the heart team meetings. Clinical data, including past medical history, medications, procedures, providers’ notes, and test results, were extracted from a web-based medical chart (Ofek, dbMotion, Pittsburg, PA, USA) shared by all Israeli hospitals and health maintenance organizations. Demographic and mortality details were verified using governmental registries.

### 2.5. Statistical Analysis

The study cohort was analyzed in its entirety and based on the occurrence of the primary outcome. Variables were reported as frequencies and percentages, medians and interquartile ranges (IQRs), or means and standard deviations. Inter-group differences were evaluated using Pearson’s chi-square, Fisher’s exact, Mann–Whitney U, or Student’s *t* tests, as suitable. Change over time in the NYHA class was assessed by the McNemar test.

To identify potential predictors for the primary outcome, a multivariable binary logistic regression analysis was constructed which incorporated baseline and procedural variables of perceived prognostic value that also possessed a *p*-value of <0.1 on univariable models.

A two-sided *p*-value of <0.05 defined statistical significance. Cases with missing data were censored from the relevant calculations. All analyses were performed using SPSS, version 24 (IBM Corporation, Armonk, NY, USA).

## 3. Results

### 3.1. Baseline Characteristics of the Study Cohort

A total of 184 patients entered the analysis and were followed for 7.0 (IQR, 3.4–12.1) years (Figure 1). The study cohort had a median age of 64 (IQR, 55–74) years and a male predominance (*n* = 141, 76.6%) (Table 1). A little more than half (*n* = 96, 53.0%) of patients presented to surgery with symptomatic HF (i.e., NYHA class II and above).

AR was mainly isolated (*n* = 134, 72.8%) and the leading AR etiology was annular dilatation (Table 2). Bicuspid AV and significant (i.e., a ≥ 4.5-cm) ascending aortic dilatation were each observed in approximately a third of cases (*n* = 60, 34.1% and *n* = 50, 29.8%, respectively). Mild-to-moderate MR or TR affected at baseline 47 (25.5%) patients, 41 (87.2%) of whom displayed only MR. The most common mitral structural anomaly was prolapse and/or flail (*n* = 44, 23.9%), followed by rheumatic disease (*n* = 14, 7.6%) and annular calcification (*n* = 8, 4.3%).

### 3.2. Procedural Aspects

Most surgeries were elective and non-urgent and involved biologic valve implantation (Table 3). Overall, 25% (*n* = 46) incorporated an ascending aortic and/or aortic root replacement and close to one fifth (*n* = 32, 17.5%) were accompanied by coronary artery bypass grafting.

### 3.3. Outcomes

The last echocardiogram, performed at 5.8 (IQR, 2.8–11.0) years after surgery, revealed the primary outcome, namely a composite of moderate or severe MR or TR, in 36 (19.6%) patients (Table 4). Concomitantly, moderate or severe MR developed in 20 (10.9%) cases, moderate or severe TR in 25 (13.6%), and both in 9 (4.9%). New-onset severe MR or TR occurred in 26 (14.1%) patients.

Resembling the preprocedural stage, the primary outcome group experienced greater functional incapacitation at one year and at the last follow-up visits (Figure 2), the latter of which proved more profound compared to the baseline (*p* = 0.044), as opposed to the non-significant difference between the baseline and last NYHA status within the no primary outcome group (*p* = 0.205). All-cause mortality rate along the entire follow-up period was also higher among patients who developed moderate or severe MR or TR (*n* = 19, 52.8% vs. *n* = 30, 20.3%, *p* < 0.001) and the risk for mortality was increased by the emergence of moderate or severe MR or TR according to univariate analysis (HR 1.78, 95% CI 1.10–3.18, *p* = 0.035). While death causes were mainly non-cardiovascular and equally distributed in the two study groups, mortality among patients sustaining the primary outcome tended to be cardiovascular more often (*n* = 8/19, 42.1% vs. *n* = 10/30, 33.3%, *p* = 0.081) (Appendix A).

### 3.4. Correlates of the Primary Outcome

Compared to patients who did not display moderate or severe MR or TR, those who did were more likely, at baseline, to exhibit atrial fibrillation/flutter, symptomatic HF, LV dysfunction, mitral valve structural abnormalities, and mild-to-moderate (vs mild or less) MR and/or TR. Also, they had a non-significantly larger ascending aortic diameter (but a marginally lower prevalence of bicuspid AV) and underwent composite graft implantation at the time of surgery more frequently. Notably, a higher incidence of moderate or severe TR, as well as of moderate or severe MR or TR, was observed among patients with mild-to-moderate (vs up-to-mild) TR or MR/TR prior to SAVR (Appendix A). The development of moderate or severe MR alone was independent of baseline MR, TR, and MR/TR severity.

Following SAVR, the primary outcome group exhibited a nominally higher residual AR grade, worse biventricular function, more pronounced chamber dilatation, and higher pulmonary arterial systolic pressure on the last documented echocardiogram (Table 4).

### 3.5. Predictors of the Primary Outcome

After multivariable analysis, four parameters were identified that independently conferred a higher risk for the emergence of moderate or severe MR or TR: the presence of atrial fibrillation/flutter (OR 3.30, 95% CI 1.10–9.85, *p* = 0.033), symptomatic HF (OR 7.42, 95% CI 3.47–14.82, *p* = 0.004), and mild-to-moderate (vs up-to-mild) MR or TR (OR 4.17, 95% CI 1.35–12.91, *p* = 0.013) preprocedure, and the use of composite graft during surgery (OR 4.20, 95% CI 1.29–13.61, *p* = 0.017) (Table 5 and Appendix A). Risk factor distribution and the probability of this composite endpoint as a function of the number of risk factors are presented in Figure 3. Interestingly, neither echocardiographic measures of chamber function and dimensions, nor aortic/mitral valve morphology or surgical aspects, were predictive of the primary outcome.

## 4. Discussion

Our study evaluated the long-term progression of non-significant MR and TR following SAVR for AR. Analyzing the data of a single-center, 184-patient cohort, the great majority (72.8%) of which displayed pure AR, we found that: 1. The primary composite outcome of moderate or severe MR or TR development occurred in about one in five cases within six years after the intervention; 2. Patients with new-onset moderate or severe MR or TR tended to exhibit a greater residual AR and were more likely to suffer biventricular dysfunction and dilatation and pulmonary hypertension on the last documented echocardiogram; 3. The emergence of moderate or severe MR or TR was associated with worse functional status and increased all-cause mortality rate during the study’s seven-year follow-up period; and 4. The risk for the development of moderate or severe MR or TR was higher in the presence of preprocedural atrial fibrillation/flutter, symptomatic HF, and above-mild MR or TR, as well as by implantation of composite graft during the index operation.

Current data regarding the course of MR and TR after AV replacement (AVR) are derived from studies that either focused on aortic stenosis (AS) or highly-selected AR cases or utilized a rather short follow-up duration. Among patients with AS, MR grade has been shown to improve overtime following both surgical [18] and transcatheter [19] AVR, while TR worsening has been observed in up to 17% of cases post-procedure, inflicting lower survival [20,21,22,23]. In patients with AR, mild MR has deteriorated in 4% of patients after SAVR according to one report [24] and moderate or severe MR has occurred in 9.4% according to another [25]. Notably, the former study spanned 3.2 years of follow-up and found a direct correlation between the follow-up time and MR progression, whereas the latter, representing 10 ± 4 years of follow-up, analyzed 97 patients, all with bicuspid AV. Our study, with its novel design, longer surveillance time, and less strict inclusion criteria therefore provides robust and real-world data on the deterioration of MR or TR following SAVR for AR, which, according to our findings, could be relevant to a non-negligible portion of patients.

Three notions may be stressed based on the study’s results. The first is that MR or TR progression post SAVR for AR is a common phenomenon associated with more advanced HF and reduced survival. Although the last documented residual AR was non-significant (i.e., up-to-moderate) in most patients, the overall AR grade (as a continuous variable) was nominally higher among those experiencing the primary outcome, suggesting a potential link between the simultaneous deterioration of the three regurgitant lesions. Whether worsening of one valvular insufficiency mediated the other or whether all the three simply represented a common underlying pathology (e.g., cardiomyopathy, connective tissue disease, or inflammatory disorder) that was not addressed by the mere AV operation, is an interesting question that could not be reliably answered by our retrospective and small-scale analysis. As for the reason accounting for the increased functional incapacitation and mortality observed among patients with MR or TR deterioration post-SAVR, our findings suggest a cardiovascular-originated mechanism. This is in view of the numerically higher cardiovascular death rate as well as the more pronounced myocardial derangement (reflected by worse ventricular function and dilatation) that accompanied MR or TR progression. Once again, and considering the study’s design, we could not determine causality, stressing the need for larger, prospective research.

The second notion arising from our work is that the development of significant MR or TR after SAVR for AR could be anticipated based on easily measurable conditions prior to the intervention. Regarding atrial fibrillation/flutter, it could be that the arrhythmic aberration partially counteracted the beneficial effect of AR correction on cardiopulmonary hemodynamics and myocardial remodeling [26,27]. Symptomatic HF and pre-existent MR or TR, on their part, might have also expressed a more profound disease state initially, as suggested by the lower LVEF observed among patients who sustained moderate or severe MR or TR post-procedure. The mechanism responsible for the association between concomitant replacement of the aorta and MR or TR progression may have been related to a more widespread disease at the outset as well or again to the presence of a shared pathology such as collagen/elastin disorders. For this matter, while aortic root and ascending aortic diameters were not independently predictive of the risk for the primary outcome per se, a nominally larger ascending aortic diameter was nevertheless noted among patients who developed moderate or severe MR or TR. Considering similarities in body habitus, general comorbidities, and immediate AR etiologies across the two study groups, this finding could imply the existence of an intrinsic under-diagnosed connective tissue-related condition(s). Importantly, pre-operative morphological (e.g., rheumatic, calcific, or degenerative) aberrations at the mitral position, although not significantly associated with the risk for the occurrence of moderate or severe MR or TR, were also more common in patients who exhibited the primary outcome, thus suggesting a role for baseline structural anomalies in MR or TR progression too, as well as supporting the possibility of an underlying common disease process. As for parameters not shown to correlate with the risk for the primary outcome, it is plausible that the study’s small sample size and low number of observations and events prevented the appreciation of additional relevant predictors, mainly specific regurgitation etiologies (rheumatic heart disease in particular [22]), RV dysfunction, and pulmonary hypertension. These may be evaluated by future larger explorations as well.

On a final and more practical note, our study underscores the importance of guideline-directed multi-modality evaluation and management of those preprocedural conditions shown to be associated with MR or TR deterioration post SAVR for AR, including atrial fibrillation, HF [28,29], and various aortopathies [30,31]. Moreover, it suggests that patients with mild-to-moderate (vs up-to-mild) MR or TR may, under certain circumstances, benefit from interventional treatment of these valvulopathies at the time of the AV surgery. While this last notion is inherently hypothetical at present and not supported by current guidelines [32,33,34,35], it should be noted that the latter are based on studies that have stemmed from different populations than ours, namely patients undergoing SAVR for AS (in case of MR correction) or mitral valve surgery altogether (in case of TR intervention). Additional, prospective trials could attest or dispute the above-mentioned impressions and help identify and validate criteria for addressing non-significant MR and TR during SAVR that is performed for AR.

### Limitations

First, the study’s single-center, retrospective design and small sample size, as well as the lack of a central and blinded data adjudication body, may all hamper the generalizability of the results. However, our cohort was one of the largest thus far in relative terms and resembled previously reported registries, therefore enhancing validity. Second, and again owing to the low number of cases and events, our predictive model should be regarded as exploratory, necessitating larger-scale confirmatory studies. Third, baseline structural characteristics of the mitral and tricuspid valves (e.g., annular dimensions) were not uniformly recorded, which prevented their consideration in the analyses. Fourth, imaging parameters were all determined by TTE studies only. However, this represented a well-accepted, real-world practice at the time of the registry, allowed for comparison of baseline and follow-up examinations in a larger subset of patients, and may facilitate the applicability of our findings. Acknowledging the fluctuating nature of regurgitant lesions, as well as the possible under-estimation of AR severity by TTE, we analyzed patients with both moderate-to-severe and severe AR at baseline.

## 5. Conclusions

In our single-center experience, significant MR or TR developed in one fifth of patients undergoing SAVR for AR by six years after the intervention, was associated with reduced functional capacity and survival, and correlated with baseline clinical and echocardiographic variables, including atrial fibrillation/flutter, symptomatic HF, mild-to-moderate MR or TR, and composite graft use. Further research is needed to validate our findings and assess their implication on the assessment and management of AR patients referred to SAVR both prior to and at the time of operation.

## Figures and Tables

**Figure 1 jcm-12-06280-f001:**
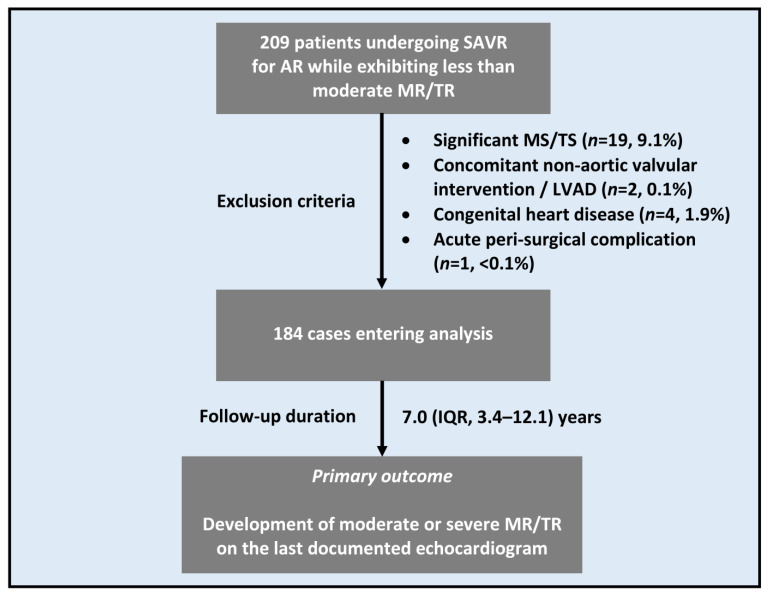
Study Flow Chart. AR = aortic regurgitation; IQR = interquartile range; LVAD = left ventricular assist device; MR = mitral regurgitation; MS = mitral stenosis; SAVR = surgical aortic valve replacement; TR = tricuspid regurgitation; TS = tricuspid stenosis.

**Figure 2 jcm-12-06280-f002:**
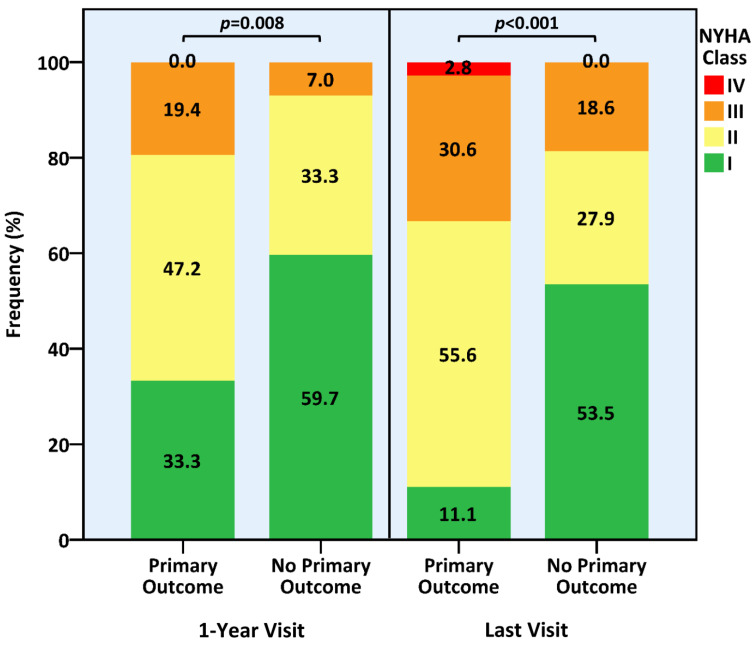
Post-Procedural Functional Status According to the Occurrence of the Primary Outcome. NYHA = New York Heart Association.

**Figure 3 jcm-12-06280-f003:**
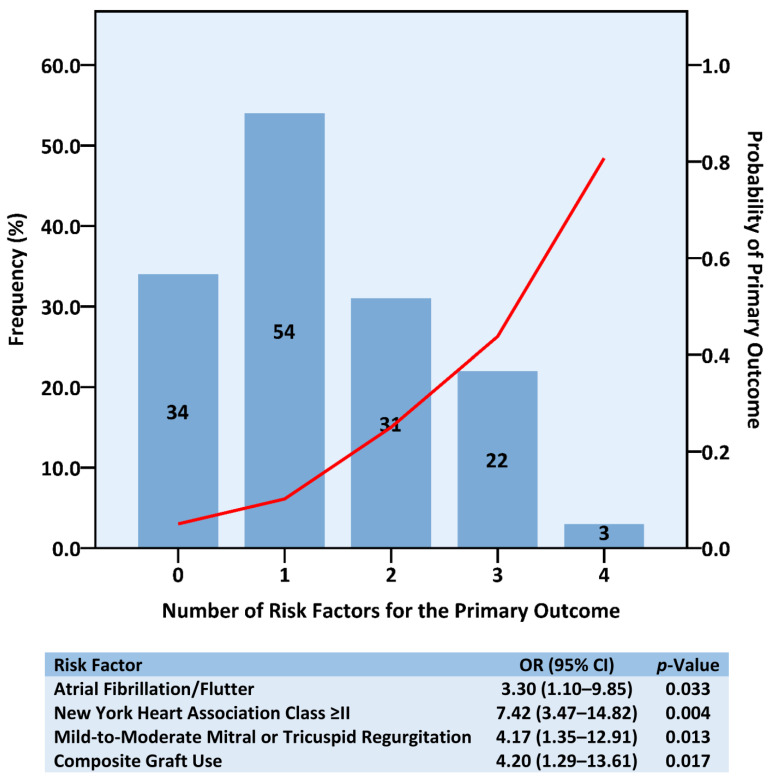
Risk Factor Burden Distribution and Correlation with the Primary Outcome. Bars represent the prevalence, at baseline, of the various risk factors counts. Red line illustrates the forecasted odds ratio for the occurrence of the primary outcome that was associated with each observed number of preprocedural risk factors. CI = confidence interval; OR = odds ratio.

**Table 1 jcm-12-06280-t001:** Baseline Clinical Characteristics.

	Total Cohort (*n* = 184)	Primary Outcome (*n* = 36)	No Primary Outcome (*n* = 148)	*p*-Value
Demographic Data				
Age				
Median (years)	64 (55–74)	70 (57–76)	62 (54–73)	0.112
≥65 years	89 (48.4)	21 (58.3)	68 (45.9)	0.182
Sex Male	141 (76.6)	24 (66.7)	117 (79.1)	0.115
Comorbidities				
Body Surface Area, Mosteller Formula (m^2^)	1.9 (1.8–2.1)	1.9 (1.7–2.0)	1.9 (1.8–2.1)	0.207
Body Mass Index (kg/m^2^)	27.8 (24.5–30.3)	28.1 (24.0–29.3)	27.6 (24.6–31.1)	0.845
Obesity	57 (33.9)	8 (24.2)	49 (36.3)	0.190
Hypertension	122 (70.1)	25 (69.4)	97 (70.3)	0.921
Diabetes Mellitus	53 (30.6)	11 (30.6)	42 (30.7)	0.991
Dyslipidemia	135 (78.0)	29 (80.6)	106 (77.4)	0.681
Smoking History	35 (20.2)	5 (13.9)	30 (21.9)	0.287
Estimated Glomerular Filtration Rate, Cockcroft Formula (mL/kg/min)	86.7 (67.9–113.5)	78.6 (60.9–112.3)	89.1 (70.1–114.2)	0.298
Stage ≥ III Chronic Kidney Disease	29 (18.8)	7 (23.3)	22 (17.7)	0.482
Ischemic Heart Disease	64 (36.8)	18 (50.0)	46 (33.3)	0.065
Prior Stroke/Transient Ischemic Attack	23 (14.6)	8 (25.0)	15 (11.9)	0.088
Atrial Fibrillation/Flutter	65 (38.2)	22 (61.1)	43 (32.1)	0.001
Cardiac Implantable Electronic Device	20 (11.8)	6 (16.7)	14 (10.5)	0.381
Marfan Syndrome	1 (0.6)	0 (0.0)	1 (0.7)	1.000
Symptomatic Status				
New York Heart Association Class				0.012
I	85 (47.0)	9 (25.0)	76 (52.4)	
II	73 (40.3)	21 (58.3)	52 (35.9)	
III	23 (12.7)	6 (16.7)	17 (11.7)	
≥II	96 (53.0)	27 (75.0)	69 (47.6)	0.005

Data are presented as number (percent) or median (interquartile range).

**Table 2 jcm-12-06280-t002:** Baseline Echocardiographic Parameters.

	Total Cohort (*n* = 184)	Primary Outcome (*n* = 36)	No Primary Outcome (*n* = 148)	*p*-Value
Study Time Prior to Surgery (days)	52 (13–192)	51 (18–182)	48 (11–208)	0.563
Aortic Valve				
Pure Aortic Regurgitation	134 (72.8)	30 (83.3)	104 (70.3)	0.114
Aortic Regurgitation Severity				0.511
Moderate-to-Severe	101 (54.9)	18 (50.0)	83 (56.1)	
Severe	83 (45.1)	18 (50.0)	65 (43.9)	
Aortic Regurgitation Etiology				0.421
Annular Dilatation	79 (65.8)	16 (64.0)	63 (66.3)	
Leaflet Prolapse/Flail	14 (11.7)	4 (16.0)	10 (10.5)	
Leaflet Restriction	10 (8.3)	3 (12.0)	7 (7.4)	
Endocarditis	15 (12.5)	1 (4.0)	14 (14.7)	
Aortic Dissection	2 (1.7)	1 (4.0)	1 (1.1)	
Moderate and Above Aortic Stenosis	54 (29.3)	6 (16.7)	48 (32.4)	0.062
Bicuspid Aortic Valve	60 (34.1)	7 (20.0)	53 (37.6)	0.049
Aorta				
Aortic Root Diameter (cm)	3.6 (3.0–4.1)	3.5 (2.9–4.3)	3.6 (3.0–4.1)	0.754
Ascending Aortic Diameter				
Median (cm)	4.1 (3.6–4.7)	4.2 (3.9–4.9)	4.0 (3.5–4.6)	0.065
≥4 cm	96 (57.1)	21 (70.0)	75 (54.3)	0.116
≥4.5 cm	50 (29.8)	11 (36.7)	39 (28.3)	0.361
Mitral and Tricuspid Valves				
Mitral Valve Anomalies				
Rheumatic Changes	14 (7.6)	3 (8.3)	11 (7.4)	0.739
Annular Dilatation	1 (0.5)	0 (0.0)	1 (0.7)	0.621
Annular Calcification	8 (4.3)	6 (16.7)	2 (1.4)	0.001
Leaflet Prolapse/Flail	44 (32.8)	13 (43.3)	31 (29.8)	0.165
Leaflet Restriction	8 (4.3)	4 (11.1)	4 (2.7)	0.048
Leaflet Tethering/Retraction	5 (2.7)	4 (11.1)	1 (0.7)	0.005
Diastolic Mitral Regurgitation	1 (0.5)	0 (0.0)	1 (0.7)	0.621
Mitral and Tricuspid Regurgitation Grade				
Mitral	0.7 ± 0.5	0.9 ± 0.3	0.6 ± 0.5	<0.001
Tricuspid	0.4 ± 0.5	0.5 ± 0.5	0.4 ± 0.5	0.164
Mild-to-Moderate Mitral or Tricuspid Regurgitation				
Mitral	41 (22.3)	15 (41.7)	26 (17.6)	0.002
Tricuspid	14 (7.7)	6 (17.1)	8 (5.4)	0.030
Either	47 (25.5)	16 (44.4)	31 (20.9)	0.004
Left Heart Chambers				
Left Ventricular Ejection Fraction				
Median (%)	60 (45–60)	50 (41–60)	60 (50–60)	0.005
<50%	52 (28.9)	16 (44.4)	36 (25.0)	0.021
Regional Wall Motion Abnormality	19 (10.3)	5 (13.9)	14 (9.5)	0.540
Left Ventricular Diastolic Dysfunction				
Any	66 (75.9)	7 (87.5)	59 (74.7)	0.673
Grade ≥2	14 (16.1)	1 (12.5)	13 (16.5)	0.772
Left Ventricular End-Systolic Diameter (cm)	3.8 (3.3–4.5)	4.1 (3.4–4.9)	3.8 (3.3–4.4)	0.213
Left Ventricular End-Diastolic Diameter (cm)	5.7 (5.2–6.3)	6.1 (5.1–6.4)	5.7 (5.2–6.1)	0.210
Left Atrial Diameter (cm)	4.2 (3.8–4.6)	4.4 (3.9–4.7)	4.1 (3.8–4.6)	0.361
Left Atrial Area (cm^2^)	23.8 (20.0–27.0)	25.5 (21.5–29.0)	23.0 (19.5–26.5)	0.072
Right Heart Chambers				
Right Ventricular Dysfunction	8 (4.5)	3 (8.8)	5 (3.5)	0.184
Right Ventricular Dilatation	3 (1.7)	1 (2.9)	2 (1.4)	0.477
Tricuspid Annular Systolic Plane Excursion (mm)	22.5 (16.8–26.8)	15.0 (13.0–16.0)	23.0 (19.0–27.0)	0.250
Pulmonary Arterial Systolic Pressure				
Median (mmHg)	27 (21–35)	32 (22–39)	26 (21–33)	0.103
>40 mmHg	8 (5.7)	4 (12.9)	4 (3.6)	0.070

Data are presented as number (percent), median (interquartile range), or mean ± standard deviation.

**Table 3 jcm-12-06280-t003:** Procedural Aspects.

	Total Cohort (*n* = 184)	Primary Outcome (*n* = 36)	No Primary Outcome (*n* = 148)	*p*-Value
Urgent Surgery	9 (4.9)	2 (5.6)	7 (4.8)	0.850
Aortic Valve Prosthesis Type				0.295
Biologic	130 (70.7)	28 (77.8)	102 (68.9)	
Mechanical	54 (29.3)	8 (22.2)	46 (31.1)	
Concomitant Aortic Vascular Intervention				
Any	46 (25.0)	11 (30.6)	35 (23.6)	0.391
Composite Graft Implantation	35 (21.6)	11 (37.9)	24 (18.0)	0.018
Concomitant Coronary Artery Bypass Grafting	32 (17.5)	8 (22.2)	24 (16.3)	0.404

Data are presented as number (percent) or median (interquartile range).

**Table 4 jcm-12-06280-t004:** Last Echocardiographic Findings.

	Total Cohort (*n* = 184)	Primary Outcome (*n* = 36)	No Primary Outcome (*n* = 148)	*p*-Value
Study Time After Surgery (years)	5.8 (2.8–11.0)	5.6 (2.7–10.8)	5.8 (2.9–11.1)	0.724
Aortic Valve				
Residual Aortic Regurgitation Severity				
Up-to-Mild	180 (97.8)	34 (94.4)	146 (98.6)	0.172
Moderate	3 (1.6)	2 (5.6)	1 (0.7)	0.098
Above-Moderate	1 (0.5)	0 (0.0)	1 (0.7)	1.000
Residual Aortic Regurgitation Grade	0.2 ± 0.5	0.4 ± 0.6	0.2 ± 0.5	0.081
Moderate and Above Aortic Stenosis	1 (0.5)	0 (0.0)	1 (0.7)	1.000
Mitral and Tricuspid Valves				
Moderate or Severe Mitral or Tricuspid Regurgitation				<0.001
Mitral	20 (10.9)	20 (55.6)	0 (0.0)	
Tricuspid	25 (13.6)	25 (69.4)	0 (0.0)	
Either	36 (19.7)	36 (100.0)	0 (0.0)	
Both	9 (4.9)	9 (25.0)	0 (0.0)	
Severe Mitral or Tricuspid Regurgitation	26 (14.1)	26 (72.2)	0 (0.0)	<0.001
Mitral Regurgitation Grade				
Median	0.8 ± 0.9	2.0 (±1.3)	0.5 ± 0.5	<0.001
Change from Baseline	0.2 ± 0.9	1.1 ± 1.2	−0.1 ± 0.6	<0.001
Tricuspid Regurgitation Grade				
Median	0.8 ± 1.0	2.3 ± 1.3	0.5 ± 0.5	<0.001
Change from Baseline	0.4 ± 1.0	1.7 ± 1.1	0.1 ± 0.6	<0.001
Left Heart Chambers				
Left Ventricular Ejection Fraction				
Median (%)	60 (50–60)	55 (31–60)	60 (50–60)	0.010
Change from Baseline (%)				
Absolute	0 (−5–5)	0 (−12–5)	0 (−4–5)	0.172
Relative	0.0 (−8.3–10.0)	0.0 (−25.0–11.9)	0.0 (−7.5–10.0)	0.150
<50%	41 (22.8)	16 (44.4)	25 (17.4)	0.001
Left Ventricular End-Systolic Diameter				
Median (cm)	3.1 (2.7–3.7)	3.4 (2.8–4.9)	3.0 (2.7–3.6)	0.011
Change from Baseline				
Absolute (cm)	−0.7 (−1.3–0.0)	−0.5 (−1.3–1.6)	−0.7 (−1.3–[−0.1])	0.097
Relative (%)	−18.7 (−30.3–0.0)	−12.5 (−31.3–12.7)	−19.4 (−30.2–[−2.7])	0.163
Left Atrial Diameter				
Median (cm)	4.3 (3.7–4.9)	5.0 (4.5–5.4)	4.2 (3.7–4.7)	<0.001
Change from Baseline				
Absolute (cm)	0.1 (−0.4–0.8)	0.8 (−0.2–1.3)	0.1 (−0.5–0.7)	0.010
Relative (%)	2.8 (−10.1–19.4)	18.6 (−3.7–30.2)	2.1 (−11.1–17.0)	0.013
Right Heart Chambers				
Right Ventricular Dysfunction	20 (11.8)	13 (40.6)	7 (5.1)	<0.001
Right Ventricular Dilatation	22 (12.9)	10 (30.3)	12 (8.7)	0.001
Pulmonary Arterial Systolic Pressure				
Median (mmHg)	29 (22–34)	35 (30–46)	26 (22–32)	<0.001
Change from Baseline				
Absolute (mmHg)	0 (−8–8)	4 (−6–18)	−1 (−9–6)	0.117
Relative (%)	−4.3 (−26.6–26.8)	10.5 (−17.2–63.6)	−8.0 (−27.3–20.3)	0.063
>40 mmHg	17 (12.7)	10 (31.3)	7 (6.9)	0.001

Data are presented as number (percent), median (interquartile range), or mean ± standard deviation.

**Table 5 jcm-12-06280-t005:** Multivariable Binary Logistic Regression Model for the Primary Outcome.

	OR (95% CI)	*p*-Value
Age (Continuous)	0.99 (0.94–1.04)	0.599
Ischemic Heart Disease	1.22 (0.41–3.61)	0.724
Prior Stroke/Transient Ischemic Attack	2.39 (0.66–8.58)	0.182
Atrial Fibrillation/Flutter	3.30 (1.10–9.85)	0.033
New York Heart Association Class ≥ II	7.42 (3.47–14.82)	0.004
Bicuspid Aortic Valve	0.37 (0.09–1.50)	0.163
Mild-to-Moderate Mitral or Tricuspid Regurgitation	4.17 (1.35–12.91)	0.013
Left Ventricular Ejection Fraction (continuous)	0.98 (0.93–1.03)	0.446
Composite Graft Use	4.20 (1.29–13.61)	0.017

CI = confidence interval; OR = odds ratio.

## Data Availability

The data underlying this article will be shared upon reasonable request to the corresponding author.

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
