# Peer review of "Progression of Non-Significant Mitral and Tricuspid Regurgitation after Surgical Aortic Valve Replacement for Aortic Regurgitation"

_jcm, 2023, doi:10.3390/jcm12196280_

Round 1

Reviewer 1 Report

The study provides valuable insights into the natural history of non-significant mitral and tricuspid regurgitation (MR and TR) following surgical aortic valve replacement (SAVR) for aortic regurgitation (AR). The retrospective analysis of 184 patients reveals important findings regarding the occurrence and impact of MR/TR after SAVR.

These findings have clinical implications, as they highlight the importance of baseline clinical and echocardiographic variables in predicting the development of significant MR/TR after SAVR. This information can aid healthcare professionals in identifying patients who may be at a higher risk of developing these complications and may require closer monitoring or alternative treatment strategies.

Some specific suggestions:

1. The Introduction section needs to be written in more detail.

2. Quality of the figures must be improved.

The language level meets the requirements for academic papers.

Author Response

Please see attached Cover Letter.

Reviewer 2 Report

The study by Kazum et al provides valuable insights into the progression of MR and TR in patients post-SAVR for AR. The findings that one-fifth of the patients developed significant MR/TR within 6 years post-SAVR and that this could be anticipated by certain baseline variables is crucial for clinicians. This can aid in better patient monitoring and early interventions. However, the retrospective nature of the study might introduce some biases, and prospective studies might be needed to confirm these findings. Overall, the research is informative and can have practical implications in the management of patients post-SAVR.

I have the a minor suggestion:

In methods it's important to state, how the echocardiographic parameters were assessed? Was it one of the authors or was this based on reports? Was there any inter observer variability? 

Author Response

Please see attached Cover Letter.

Reviewer 3 Report

Shirit Kazumet al. present an original article about the natural history of non-significant mitral and tricuspid regurgitation following surgical aortic valve replacement for aortic regurgitation. Although the topic is interesting and the manuscript well-written, some considerations need to be clarified.

1. Introduction: In the introduction is lacking how to diagnose severe aortic valve regurgitation. In addition, multimodality imaging approach is key to the diagnosis, thus describe the pathway and algorithm for diagnosing with different techniques (such as echocardiography, CT and CMR).

Please, add the following references to improve the quality of the new sentences:
- Lancellotti P, et al; Scientific Document Committee of the European Association of Cardiovascular Imaging. Multi-modality imaging assessment of native valvular regurgitation: an EACVI and ESC council of valvular heart disease position paper. Eur Heart J Cardiovasc Imaging. 2022 Apr 18;23(5):e171-e232. doi: 10.1093/ehjci/jeab253..
- Siani A, et al. Aortic regurgitation: A multimodality approach. J Clin Ultrasound. 2022 Oct;50(8):1041-1050. doi: 10.1002/jcu.23299.

2. Discussion: The development of symptoms during deterioration of MR and TR post SAVR should be discussed, especially the role of NYHA class and its utility. Please add concepts on this interesting point, the number of patients with HF symptoms during the follow-up, and possible future research study directions.

Please, add the following references to improve the quality of the new sentences:
- Authors/Task Force Members:; McDonagh TA, et al. ESC Scientific Document Group. 2021 ESC Guidelines for the diagnosis and treatment of acute and chronic heart failure: Developed by the Task Force for the diagnosis and treatment of acute and chronic heart failure of the European Society of Cardiology (ESC). With the special contribution of the Heart Failure Association (HFA) of the ESC. Eur J Heart Fail. 2022 Jan;24(1):4-131. doi: 10.1002/ejhf.2333.
- Perone F, et al. An Overview of Sport Participation and Exercise Prescription in Mitral Valve Disease. J Cardiovasc Dev Dis. 2023 Jul 18;10(7):304. doi: 10.3390/jcdd10070304.

Author Response

Please see attached Cover Letter.

Round 2

Reviewer 3 Report

The authors have satisfactorily responded to my comments and doubts.
Congratulations. 

Author Response

Please see attached Cover Letter.